# Molecular Detection of Multiple Genotypes of *Orientia tsutsugamushi* Causing Scrub Typhus in Febrile Patients from Theni District, South India

**DOI:** 10.3390/tropicalmed8030174

**Published:** 2023-03-16

**Authors:** Krishnamoorthy Nallan, Gopinathan Rajan, Lallitha Sivathanu, Panneer Devaraju, Balaji Thiruppathi, Ashwani Kumar, Paramasivan Rajaiah

**Affiliations:** 1ICMR-Vector Control Research Centre, Field Unit, Madurai 625 002, India; 2Department of Microbiology, Government Medical College and Hospital, Theni 625 512, India; 3ICMR-Vector Control Research Centre, Puducherry 605 006, India

**Keywords:** scrub typhus, *Orientia tsutsugamushi*, antigenic variants, chigger, Karp, Kato, Kawasaki, India

## Abstract

Scrub typhus (St) is a re-emerging mite-transmitted public health problem in Southeast Asia with escalating case incidences in the endemic areas. Though, more than 40 genotypes of the causative agent *Orientia tsutsugamushi* (*Ot*) have been documented, the information on the circulating genotypes in India is scanty. A hospital-based retrospective screening was undertaken to map the circulating molecular subtypes of the etiological agent in serologically confirmed scrub typhus (St) human cases, by targeting the GroEL gene of *O. tsutsugamushi* using the nested polymerase chain reaction method. Nine out of 34 samples (26%) yielded positive results and DNA sequencing analysis of six positive samples out of nine revealed that the sequences were related to three major genotypes, such as Karp (HSB1, FAR1), Kato (Wuj/2014, UT76), and Kawasaki (Kuroki, Boryong, Gilliam, and Hwasung). Additionally, the St-positive samples exhibited 100% and 99.45%; 97.53% and 97.81%; 96.99% nucleotide identity with the closely related Karp, Kato, and Kawasaki-related sequences, respectively. Overall, 94% of the nucleotides were conserved, and the variable site was 20/365 (5.5%). The prevalence of multiple genotypes among human cases further stresses the need to conduct in-depth studies to map the genotypes and their clinical relevance, and the contributing risk factors for the emergence of St cases in this area.

## 1. Introduction

Scrub typhus (St) is a re-emerging rickettsial infection caused by an obligate intracellular bacterial pathogen, *Orientia tsutsugamushi (Ot)*, transmitted through the bite of chigger mites belonging to the genus *Leptotrombidium*. The pathogen is maintained in nature predominantly by rodents and mites, and these mites are prevalent in dense scrub vegetation and bushes. Though the distribution of the etiological agent was believed to be restricted to the “*tsutsugamushi *triangle” [1,2], its expanded distribution in Chile and Saudi Arabia indicates its widespread occurrence and public health importance worldwide [3,4]. 

Scrub typhus is endemic in India and about 40 outbreaks of St cases were recorded during 2008–2017, from almost all the states of the country [5]. Worldwide, more than 40 serotypes with higher antigenic variation have been recorded, which may be expected to hamper the diagnostic specificity [6,7]. Though scrub typhus remains to be a serious public health problem in the state of Tamil Nadu [8,9,10] the available data on the circulating genotypes is very limited, in particular, the molecular genotyping of *Ot*. In Tamil Nadu, the highest numbers of St cases were reported, particularly during the cooler months between September to January, every year [11]. Apart from Tamil Nadu, scrub typhus has also been reported among children and adults in the neighboring state of Kerala in recent times [12]. Based on the higher incidence of cases during the cooler months, the adjacent areas of Madurai, coastal areas in Tamil Nadu, and the northeast of Andhra Pradesh have been identified as “hotspots” of St [11]. Theni district shown in Figure 1, also known as “Cumbum Valley,” is located at the foot of the Western Ghats (bordering Kerala State), making it an ideal niche for St transmission. It consists of approximately 80 villages, with 47 percent of the population mainly involved in agriculture and allied activities [13]. The “Cumbum Valley” experiences a cool climate throughout the year; the temperatures vary from as low as 4 °C to 25 °C, and receive rain in the Southwest and Northeast monsoon. Nonetheless, temperature, rainfall, and other ecological and socioeconomic factors contribute to disease transmission, but the true disease burden globally is underreported irrespective of the region due to delayed diagnosis [1].

Hence, a retrospective pilot attempt was made to detect the circulating genotypes of *Ot* in human cases admitted to the tertiary care medical centre at the Government Medical College and Hospital, Theni, Tamil Nadu. This pilot study highlights the distribution of diverse genotypes of *Ot* associated with confirmed human cases of St that are arising from this area.

## 2. Materials and Methods

### 2.1. Sample Collection

Patients who visited Government Medical College and Hospital seeking treatment with clinical symptoms including acute fever, breathlessness, cough, myalgia, headache, chills, dry cough, lymphadenopathy, and gastrointestinal disturbances were included in this study. Serum samples of these patients were screened for anti-scrub typhus antibodies using IgM antibody ELISA kit (Scrub Typhus Detect IgM ELISA, InBios International, Inc., Seattle, WA, USA). The samples that tested positive (*n* = 34) for St were subjected to molecular genotyping analysis. After the sample collection, all the patients with the above symptoms were treated empirically with tetracycline. The Institutional Human Ethics Committee has approved this retrospective study. The positive serum samples were transported to the laboratory at the Vector Control Research Centre, Filed Unit, Madurai by maintaining the cold chain. DNA was extracted from the serum samples using the commercially available QIAamp DNA Blood Mini Kit. (Qiagen, Germany).

### 2.2. Molecular Diagnosis

A nested PCR assay targeting the *Rickettsia* GroEL gene was carried out to detect the *Ot* using the primers Gro-1, 5′-AAGAAGGA/CGTGATAAC-3′ and Gro-2, 5′-ACTTCA/CGTAGCACC-3′ and TF1, 5′-ATATATCACAGTACTTTGCAAC-3′ and TR2, and 5′-GTTCCTAACTTAGATGTATCAT-3′, respectively to amplify a 365 bp sequence as described by Weihong Li et al. [14]. The amplified DNA fragment was custom-sequenced using commercial sequencing services. The sequences obtained were analyzed using the software chromas ver. 2.6.6, Technelysium, DNA Sequencing Software (South Brisbane, QLD, Australia) and the DNA sequences were subjected to BLAST analysis to check their similarity with the GenBank database. The sequences that were closest neighbors to the query sequences were used for further phylogenetic analysis in MEGA 7 Molecular Evolutionary Genetics Analysis software [15] to confirm the *Ot* genotypes. For estimating genetic distances, a neighbor-joining tree was constructed using the Kimura 2-parameter [K2P] model with 500 bootstrap replicates. Following phylogenetic analysis, six DNA sequences were divided into three groups based on phylogenetic clades to calculate inter and intra genotypic differences.

## 3. Results

Out of 34 samples subjected to PCR analysis, nine samples (26%) yielded *Ot*-specific amplicons. The geographical locations and details of the PCR results are shown in Figure 1 and Table 1. The PCR-positive samples were collected from the areas of Aundipatti, Chinnamanur, Theni, and Uthamapalyam. Six sequences of the groEL gene were obtained in this study (Table 1). The NCBI BLAST analysis revealed that they belong to three strains. Two sequences were genetically identical (ON156000, ON156001) and one was nearly identical (ON155999) to the Karp strain reported in Japan, the UK, and the USA. Two sequences were genetically similar (ON156002, ON156003) to the Kato strain reported in China and the UK with sequence similarity of 97.53% and 97.81%, respectively. Another sequence (ON156004) is genetically similar to the Kawasaki strain reported in Korea, Japan, and the UK with a sequence similarity of 96.99% (Table 1).

Phylogenetic analysis revealed that sequences ON156000, ON156001, and ON155999 belong to a clade with Karp-related sequences reported in Japan, the UK, and the USA. However, the bootstrap support for this clade is low (<50%) (Figure 2). The sequences ON156002 and ON156003 although belong to the Kato-related clade with high bootstrap supported (93%) these two sequences obtained in the present study formed a distinct sub-clade from specimens of the Kato strain reported in China and UK. Similarly, a sequence ON156004 although belonging to a clade of the Kawasaki related to those reported in the UK, Korea, and Japan, this sequence is formed a distinct lineage (Figure 2).

The number of base substitutions per site from mean diversity calculations for the entire population (*d)*, and the number of base substitutions per site from averaging over all sequence pairs within each group and between groups are shown in Table 2.

The average number of base substitutions per site within and between Kawasaki-related groups was 0.03, 0.05. The nucleotide sequence homology of the single Kawasaki (ON156004 VCRC) with related sequences, viz., Kuroki-Boryong-Kawasaski; Hawsung, and Gilliam was found to be 96.9%, 96.7%, and 96.4%, respectively. In the overall analysis, 94.5% of the nucleotides were found to be conserved, and 5.5% remained variable, i.e., 20/365 nucleotide substitutions were found when compared with the Kawasaki-related sequences in the phylogenic clade. Furthermore, the nucleotide substitution rate was estimated to be 12/365 (3.3%) between ON156004 VCRC—Kuroki-Boryong-Kawasaski and 3.6%, with Hwasung-Gilliam—ON156004 VCRC.

Three Karp-like sequences were further divided into two groups viz, ON156000-ON156001, and ON155999-LS398548, since two sequences were genetically identical (ON156000, ON156001) and one was nearly identical (ON155999) to the Karp strain, and the pairwise distance was calculated with the homologous LS398548-Karp (UK) sequence, which exhibited 100% query coverage and 99.45% identity with ON155999. The estimated pair-wise distance was 0.06 and 0.003 for ON155999 and ON156000- ON156001, respectively. The other two sequences (ON156002 and ON156003) were found aligned closely with Kato-related genotypes (LS398550, LS398552, and CP044031) and LS398550 (UK). The overall nucleotide mean distance to Kato-related ON156002 and ON156003 was found to be 0.02, which evidently demonstrates less diversity than the Kawasaki strain (Figure 2).

## 4. Discussion

Scrub typhus (St) is a reemerging public health problem in India, with increasing case incidences across the country [16,17,18]. During the last 10 years, an alarming number (18,781) of confirmed St cases was recorded in southern India [16]. Despite the fact that St is distributed around the country, the data on the circulating genotypes of *Ot* need to be determined, as it is important for designing region-wise diagnostic kits, understanding serotype-specific clinical severity among humans, and designing future vaccine strategies. 

The present pilot study has for the first time recorded data on the circulating genotypes of *Ot* in Theni districts in Cumbam Valley, Tamil Nadu. The preliminary study has detected the circulation of the Karp, Kato and Kawasaki-related genotypes in humans. The detection of molecular evidence of *Ot* in human clinical samples clearly indicates the widespread occurrence and circulation of multiple genotypes causing human infections in this area.

Compared with Karp-related sequences, the DNA sequence analysis with closely related sequences indicates that Kato and Kawasaki exhibited a high nucleotide diversity (d) of 0.02 and 0.03, respectively. Karp-related sequences also showed different genetic patterns among them. Interestingly, samples positive for Karp-related serotypes were collected from two areas, namely, Chinnamanur and Sitharpatti, which are situated 31 km apart. The genetic variation (0.06) in nucleotide sequence between ON155600-ON1556001 and ON155999 the same Karp-related genotype denotes the event of multiple introductions and intragenic recombination of genotypes in this region and their establishment in pockets [19,20]. The higher within-strain mean distance of Kato and Kawasaki, especially the second, needs further in-depth study to delineate its serotypes in this area. The 97.5% and 97.8% identities with Kato sequences and the 96.9% identity with Kawasaki and Boryong-related sequences suggested the presence of novel variants in this area. Nevertheless, the *Ot* genotypes detected in these areas were aligned with three clades, and all the sequences form sub-clades except ON155600, and ON155601 Karp-related sequences. These results also indicate that *Ot* circulating in rural and urban area are different because only Kato-related strain was detected in Theni urban. It is obvious to note that no DNA sequences in GenBank were 100% identical to our Kato and Kawasaki-related sequences. However, only two Karp-related sequences (ON156000-VCRC and ON156001-VCRC) showed 100% query coverage and percentage identity with the *Ot* GroEL DNA sequence from Japan. The higher sequence homology we obtained with the *Ot* sequence submitted from S. Korea, China, Japan, and the United Kingdom could possibly explain the trafficking of pathogens from this country to southern India, probably through migratory birds [21,22], but needs further confirmation. 

The Kawasaki-related genotypes, in particular, showed 20/365 (5.5%) substitution with 96.9% identity to the Kuroki, Kawasaki, and Boryong strains. The nucleotide identity with Hwasung and Gilliam was 96.7% and 96.4%, respectively, which shows that they are genetically more distant than the Karp and Kato-like serotypes circulating in this area. DNA sequences generated in other studies using the same primers from South India and that are available in the NCBI database were matched with our sequences. However, due to the low percentage (65–70%) of query coverage and identity, we could not include it in the analysis.

The pilot data also indicate that St remains an important clinical entity among the pyrexia of unknown origin (PUO) cases arising in these areas. Observation of rich nucleotide diversity among the genotypes detected indicates the presence of novel variants in independent foci and the possible establishment of hotspots in the affected area [20]. Antigenic diversity in *O. tsutsugamushi* indicates that the pathogen is continuously evolving through the genetic recombination events among the locally circulating genotypes [19]. Additionally, the detection of closely related genotypes with nucleotide diversity demonstrates the ongoing event of the independent introduction of pathogens in this region and their establishment as endemic pockets. Future studies assessing the serological assays that are in use in this region are highly warranted, as it has been shown that the circulation of multiple serotypes hampers diagnostic efforts during surveillance [6,7]. In-depth studies mapping the diversity of the pathogen could complement the efforts of vaccine development attempts.

## 5. Conclusions

In India, the prevalence of all major genotypes of Karp, Kato, Kawasaki, and Gilliam has been recorded [18]. The increasing list of novel genotypes around the endemic areas means that in-depth studies to map the diversity and their clinical relevance in India are recommended. The present study documents the occurrence of the rich diversity of *O. tsutsugamushi* and reminds us that scrub typhus is one of the important clinical entities among PUO cases. Since Theni district is located at the foot of the Western Ghats, sharing the border district of Kerala, the existence of suitable ecological and behavioral factors in this valley might favorably support the transmission of *Ot* to the local population through the rodent–mites cycle. The independent introduction of pathogens in these areas with the genetic evidence indicates the silent establishment of pathogens in hotspots and the possible emergence of recombinant genotypes and their noteworthy impact on future surveillance activities.

## Figures and Tables

**Figure 1 tropicalmed-08-00174-f001:**
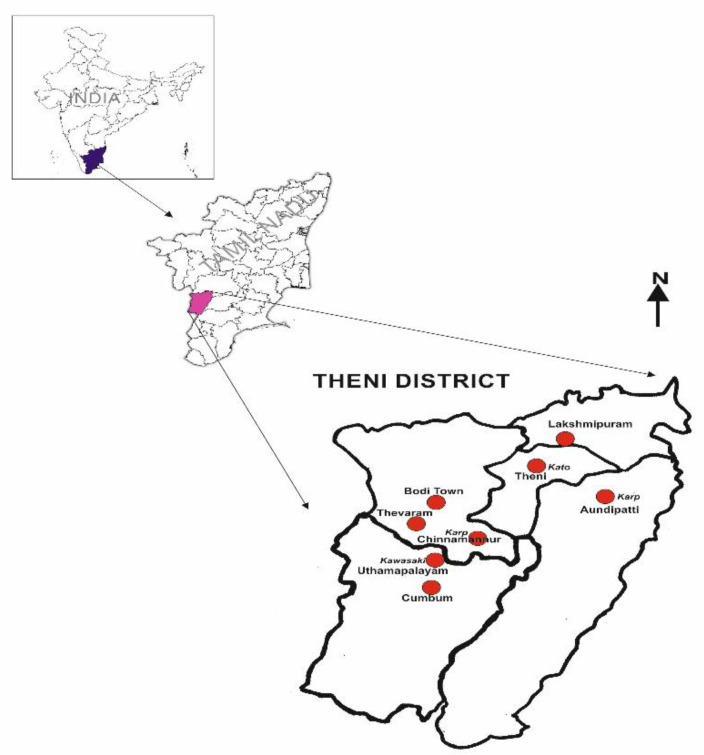
Location of the study area and *O. tsutsugmushi* genotype detected in this study in Theni district, Tamil Nadu.

**Figure 2 tropicalmed-08-00174-f002:**
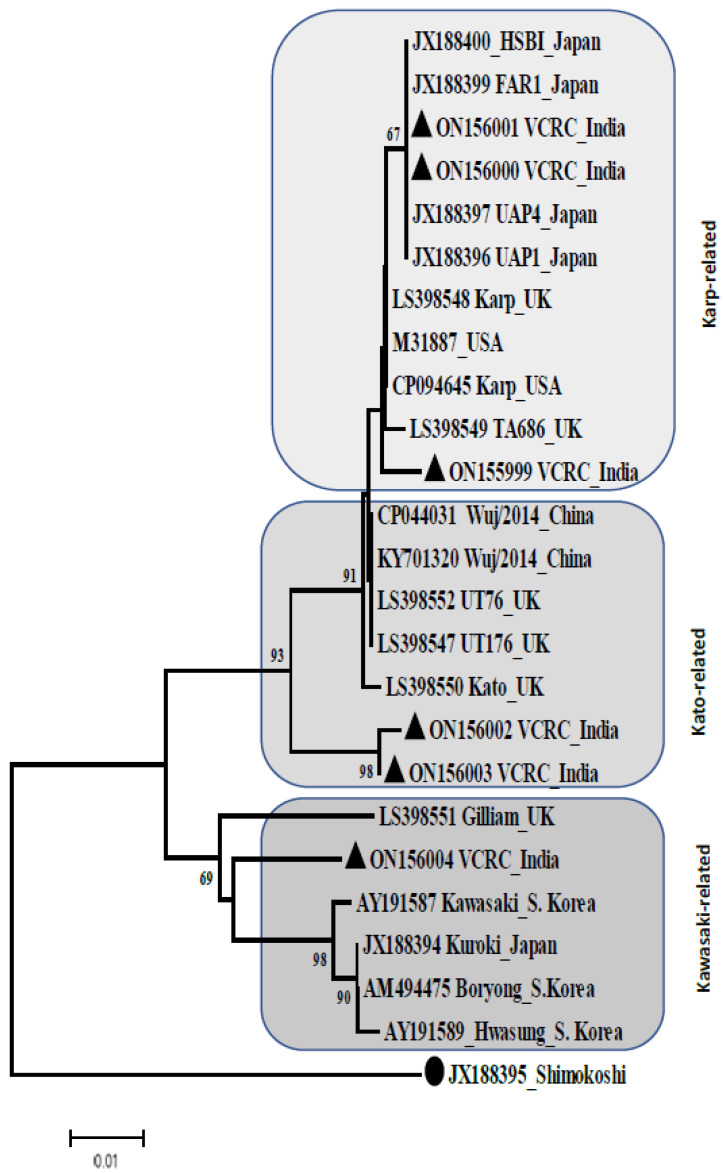
Phylogenetic analysis of *O. tsutsugamushi* using the *GroEL* gene sequences (365 bp) from Karp, Kato, and Kawasaki-related serotypes. The neighbor joining (NJ) tree was generated using closely related sequences with 500 bootstrap replicates. ▲: Sequences from this study; ●: out-group.

**Table 1 tropicalmed-08-00174-t001:** The sample details and GenBank accession numbers of the *O. tsutsugamushi* detected in this study.

Sl No	Sample ID	Age/ Sex	Locality	GenBank Acc No. (VCRC)	Related Strains	Closest GenBank Acc No.	Percentage Identity
1	ST10	41/M	Sithrapatti- Aundipatti	ON155999	Karp, UT76	LS398548	99.45%
2	ST18	47/M	Chinnamanur	ON156000	HSBI, Karp	JX188400	100%
3	ST21	34/F	Chinnamanur	ON156001	HSBI, Karp	JX188400	100%
4	ST23	45/F	Theni	ON156002	Wuj/2014, Kato	CP044031	97.53%
5	ST26	55/M	Theni	ON156003	Wuj/2014, Kato	CP044031	97.81%
6	ST27	63/F	Uthamapalayam	ON156004	Kuroki, Kawasaki	AY191587, JX188394	96.99%

**Table 2 tropicalmed-08-00174-t002:** Analysis of mean distance within and between *O. tsutsugamushi* sequences.

*Ot* Groups	Within Group Mean Distance		Between Group Mean Distance		
	Distance (d)	Standard error (*s.e*)	Distance (d)	Standard Error (*s.e*)	
1. Karp	0.00	0.00		0.01	0.01
2. Kato	0.02	0.01	0.02		0.01
3. Kawasaki	0.03	0.01	0.05	0.05	

Estimate of the mean evolutionary diversity for the entire population (d) = 0.03 (s.e 0.01).

## Data Availability

Not applicable.

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
