# Peer review of "Molecular Detection of Multiple Genotypes of Orientia tsutsugamushi Causing Scrub Typhus in Febrile Patients from Theni District, South India"

_tropicalmed, 2023, doi:10.3390/tropicalmed8030174_

Round 1

Reviewer 1 Report

This manuscript uses molecular techniques to determine Orientia tsutsugamushi in patients with scrub typhus in southern India. Overall, the results are interesting and will be useful for further monitoring and examination of this important pathogen. Following are my comments and suggestions.

1. In Lines 32 and 33, the names of OT and Leptotrombidium are in different fonts from other texts. Please correct this. Please also check throughout the manuscript and correct it.

2. In Line 88, more details of the phylogenetic analysis are requiring e.g. which method, and parameter setting.

3. Results section, in my opinion, the present version of the Results section is too complicated and difficult to follow therefore, I suggest revising this section. I recommend the following way to report:

“Six sequences of the groEL gene were obtained in this study (Table 1). The NCBI BLAST analysis revealed that they belong to three strains. Two sequences were genetically identical (ON156000, ON156001) and one was nearly identical (ON155999) to the Karp strain (Table 1) reported from Japan, the UK, and the USA. Two sequences were genetically similar (ON156002, ON156003) to the Kato strain reported from China and the UK with sequence similarity of 97.53% and 97.81%, respectively. Another sequence (ON156004) is genetically similar to the Kawasaki strain reported from Korea, Japan, and the UK with a sequence similarity of 96.99% (Table 1).

Phylogenetic analysis revealed that sequences ON156000, ON156001, and ON155999 belong to a clade with Karp-related sequences reported from Japan, the UK, and the USA (Fig. 1). However, the bootstrap support for this clade is low (<50%) (Fig. 1). The sequences ON156002 and ON156003 although belong to the Kato-related clade with high bootstrap supported (93%) these two sequences obtained in the present study formed a distinct subclade from specimens of the Kato strain reported from China and UK. Similarly, a sequence ON156004 although belonging to a clade of the Kawasaki related to those reported from the UK, Korea, and Japan, this sequence is formed a distinct lineage (Fig. 1).”

4. In Figure 1, the bootstrap value lower than 50% should not show in the figure.

5. Discussion section, I suggest that the author should consider the following points for discussion:

- Level of genetic similarity and results of phylogenetic analysis e.g. (1) three sequences of the Karp-related strain was genetically identical or very similar to previously reported from several countries, why? (2) two sequences from India that belong to the Kato-related clade are genetically distinct from other sequences of this strain reported previously, why? (3) A sequence from India obtained in the present study is genetically close to the Kawasaki strain but with a considerable level of genetic divergence (>3%), why?

Author Response

Dear sir,

Thank you very much for the valuable comments. Please find attached herewith the reply to the comment on our manuscript. We have meticulously attended to all the queries and clarified them point by point. In this connection, if any more clarification is required, please let us know.

Thanking you.

With Kind regards

Krishnamoorthy. N

Reviewer 2 Report

The authors aimed to detect the circulating genotypes of Orientia tsutsugamushi in human cases admitted to a tertiary care hospital in India. 

Although the study sounds good, the number of samples is very limited. Moreover, the tools used in the experiments can significantly be improved.

The current findings (only 6 isolates) do not allow such conclusions. The authors should increase the sample size to draw strong conclusions.

I also suggest combining figures 1 and 2. Table 2 can be move to supplementary data.

Author Response

(The authors gave the same response as above.)

Reviewer 3 Report

1. Some grammatical errors, Orientia tsutsugamushi and GroEL should be italicized, please check throughout the manuscript

2. line 31, "rickettsial infection", not rickettsial

3. lines 90-94 should be shown in the results section

4. line 122, "substitution rate", please revise

5. Figure 3, the figure should be shown with high resolution

6. lines 165 and 184, "ongoing evolution", not ongoing evolution, should be novel variants.

Author Response

(The authors gave the same response as above.)

Reviewer 4 Report

Introduction

Line 32 and 33 the format of name of organism and mite should be corrected

The authors should add information about the signs, diagnosis and epidemiology of the disease either in the world or in India

Materials and methods

How author determine the positive patient with ELISA

How determine the sample size of the study

Why did you chose only six positive sample for sequencing?

Results

Please, improve the resolution of the figure 3

Author Response

(The authors gave the same response as above.)

Round 2

Reviewer 2 Report

The authors did not modify the manuscript according to my suggestions.

Unfortunately, my negative decision is still the same.

Author Response

Thank you for your valuable comments, We have carried out all the corrections as suggested.
